# Positive Effects of Ger-Gen-Chyn-Lian-Tang on Cholestatic Liver Fibrosis in Bile Duct Ligation-Challenged Mice

**DOI:** 10.3390/ijms20174181

**Published:** 2019-08-26

**Authors:** Zi-Yu Chang, Chin-Chang Chen, Hsuan-Miao Liu, Yuan-Chieh Yeh, Tung-Yi Lin, Tzung-Yan Lee, Tse-Hung Huang

**Affiliations:** 1Department of Traditional Chinese Medicine, Chang Gung Memorial Hospital, Keelung 204, Taiwan; 2Institute of Traditional Medicine, School of Medicine, National Yang-Ming University, Taipei 112, Taiwan; 3Research Center for Chinese Herbal Medicine, Chang Gung University of Science and Technology, Taoyuan 333, Taiwan; 4Graduate Institute of Traditional Chinese Medicine, Chang Gung University, Taoyuan 333, Taiwan; 5School of Traditional Chinese Medicine, Chang Gung University, Taoyuan 333, Taiwan; 6School of Nursing, National Taipei University of Nursing and Health Sciences, Taipei 112, Taiwan; 7Graduate Institute of Health Industry Technology, Chang Gung University of Science and Technology, Taoyuan 333, Taiwan

**Keywords:** Ger-Gen-Chyn-Lian-Tang, liver fibrosis, HIF-1α, oxidative stress, angiogenesis

## Abstract

The purpose of this study was to investigate whether Ger-Gen-Chyn-Lian-Tang (GGCLT) suppresses oxidative stress, inflammation, and angiogenesis during experimental liver fibrosis through the hypoxia-inducible factor-1α (HIF-1α)-mediated pathway. Male C57BL/6 mice were randomly assigned to a sham-control or bile duct ligation (BDL) group with or without treatment with GGCLT at 30, 100, and 300 mg/kg. Plasma alanine aminotransferase (ALT) levels were analyzed using a diagnostic kit. Liver histopathology and hepatic status parameters were measured. Compared to control mice, the BDL mice exhibited an enlargement in liver HIF-1α levels, which was suppressed by 100 and 300 mg/kg GGCLT treatments (control: BDL: BDL + GGCLT-100: BDL + GGCLT-300 = 0.95 ± 0.07: 1.95 ± 0.12: 1.43 ± 0.05: 1.12 ± 0.10 fold; *p* < 0.05). GGCLT restrained the induction of hepatic hydroxyproline and malondialdehyde levels in the mice challenged with BDL, further increasing the hepatic glutathione levels. Furthermore, in response to increased hepatic inflammation and fibrogenesis, significant levels of ALT, nuclear factor kappa B, transforming growth factor-β, α-smooth muscle actin, matrix metalloproteinase-2 (MMP-2), MMP-9, and procollagen-III were found in BDL mice, which were attenuated with GGCLT. In addition, GGCLT reduced the induction of angiogenesis in the liver after BDL by inhibiting vascular endothelial growth factor (VEGF) and VEGF receptors 1 and 2. In conclusion, the anti-liver fibrosis effect of GGCLT, which suppresses hepatic oxidative stress and angiogenesis, may be dependent on an HIF-1α-mediated pathway.

## 1. Introduction

Extracellular matrix (ECM) degradation, which disrupts the normal architecture of the liver, is a major characteristic in hepatic fibrosis, a common pathway to chronic liver disease [1,2]. It has been reported that fibrogenesis-mediated impaired blood supply induces a hypoxic microenvironment as a consequence of fibrosis, which leads to liver cirrhosis [3]. Previous evidence suggests that hypoxia promotes inflammation and fibrosis in the liver by the activation of hypoxia-inducible factor-1α (HIF-1α) and nuclear factor kappa B (NF-κB) under the regulation of fibrogenesis and angiogenesis [4,5]. Furthermore, hypoxic-induced generation of reactive oxygen species greatly aggravates inflammation [4], which promotes ECM deposition [6]. In addition, activated hepatic stellate cells (HSCs) and portal myofibroblasts stimulate angiogenesis by inducing vascular endothelial growth factor (VEGF) in the pathogenesis of liver fibrosis [7,8]. The HIF-1 protein complex is activated in response to a hypoxic microenvironment and acts as a vital transcriptional regulator by binding to nuclear hypoxia response elements. It is noteworthy that hypoxia contributes to organ fibrosis by causing epithelial-to-mesenchymal transition (EMT) in tubular epithelial cells via HIF-1 signaling [9]. Moreover, the transforming growth factor-β (TGF-β) signal pathway also plays a vital role in HSC activation through the stimulation of HIF-1α [10,11].

Bile duct ligation (BDL) is a surgical procedure in the common bile duct that is used to cause extrahepatic biliary obstruction, which mimics cholestatic liver damage. The process of BDL-induced liver cirrhosis is that it induces the rapid growth of biliary epithelial cells and oval cells, then causes bile ductular proliferation, cholestasis, and fibrosis, ultimately leading to secondary biliary cirrhosis [12,13,14]. Following this process, rodent models exhibit alterations of phagocytic, immune, and macrophage function, as well as neutrophil recruitment in the liver tissues [15,16,17]. In this regard, BDL in mice is a widespread model used to induce biliary inflammation, fibrosis and cholestatic liver injury [18]. In addition, persinusoidal and periportal fibrosis have both been reported to have fully developed on day 10 and day 20, respectively, after BDL surgery in mice [19]. Previously, Gabbiani and his colleagues suggested that a marked expansion of portal fibroblasts played an initiating role in the early development of liver fibrosis by rapidly inducing α-smooth muscle actin (α-SMA) expression and ECM deposition in BDL-challenged rats [20,21]. Furthermore, Moczydlowska et al. demonstrated that matrix metalloproteinase-2 (MMP-2) and MMP-9 were highly regulated and involved in the mechanism of liver fibrosis via upregulation of HIF-1α and NF-κB transcriptional activation after BDL challenge [22]. A previous study indicated that HIF-1α is activated in the livers of BDL-treated mice [23]. Of note, a large amount of nuclear HIF-1α accumulated in bile obstruction- and periportal region-adjacent hepatocytes and macrophages in the early stage of BDL. Moreover, hypoxia induced the expression of VEGF in an in vitro study of rodent hepatocytes [24,25], and VEGF has been shown to stimulate the proliferation and chemoattraction of HSCs to promote liver fibrosis [26,27,28]. Additionally, both receptors between integrins and TGF-β induce signaling that stimulate collagen biosynthesis and fibrotic generation [29].

Ger-Gen-Chyn-Lian-Tang (GGCLT), a standardized Chinese herbal medicine used in Taiwan, contains *Puerariae radix*, *Scutellariae radix*, *Coptidis rhizome*, and *Glycyrrhizae radix.* A previous study revealed that GGCLT treatment reduced progression of atherosclerosis in an apolipoprotein E^-/-^ mouse model [30]. Meanwhile, our previous report suggested that GGCLT protected hepatic cells from thioacetamide-induced liver injury via the reduction of oxidative stress status [31]. It has been reported that puerarin, baicalin, berberine and glycyrrhiza are the effective constituents of *Puerariae radix*, *Scutellariae radix*, *Coptisis rhizoma* and *Glycyrrhizae radix*, respectively [32,33,34,35]. Previous studies have indicated that puerarin, baicalin, and berberine can suppress cell proliferation and angiogenesis in diabetic retinopathy, pulmonary hypertension and tumor-directed capillary formation, respectively, by downregulating HIF-1 expression [36,37,38]. A better understanding of the cellular and molecular mechanisms in liver fibrogenesis can be gained by using multiple experimental rodent model systems. Therefore, we investigated whether GGCLT suppresses the oxidative stress, inflammation, and angiogenesis underlying hypoxia during BDL-induced liver fibrosis in mice. 

## 2. Results

### 2.1. GGCLT Treatment Effect on Liver Injury and Hepatic Oxidative Stress

At first, the four main index ingredients of our prepared GGCLT, specifically puerarin, baicalin, berberine and glycyrrhiza, were identified using high-performance liquid chromatography (HPLC), and are shown in Figure 1. As shown in Figure 2A, a large deposition of collagen fibers in perivenular and periportal areas of the liver lobules was found in BDL-challenged mice. Subsequently, mice that received 300 mg/kg GGCLT showed a thinner septum of collagen, a decrease in inflammatory cells, and retained hepatic parenchyma more than the mice with BDL. In comparison with the sham-control group, a significant elevation of plasma alanine aminotransferase (ALT) levels and hepatic HIF-1α mRNA expression occurred in the BDL mice, which was suppressed by GGCLT treatment, especially at doses of 100 and 300 mg/kg (Figure 2B,C). Moreover, the hydroxyproline and thiobarbituric acid reactive substances (TBARS) levels in the liver tissues were markedly increased in the BDL mice in comparison with the sham-control mice. Administration of GGCLT significantly attenuated the increase in hydroxyproline and TBARS in the BDL mice (Figure 3A,B). In addition, a significant decrease in hepatic glutathione (GSH) activity occurred after the BDL challenge; this effect was reversed by treatment with GGCLT at doses of 100 and 300 mg/kg (Figure 3C).

### 2.2. GGCLT Treatment Effect on Fibrogenesis and Angiogenesis-Related Factors in the Liver

The results of Western blotting showed that hepatic levels of NF-κB, α-SMA, TGFβ-R1, and TGFβ-R2, four fibrogenesis-related factors, were obviously elevated to a greater extent in the BDL mice than in the sham-control mice. However, these effects were suppressed by 300 mg/kg GGCLT administration (Figure 4). In parallel, the hepatic mRNA levels of TGF-β, MMP-2, MMP-9, and procollagen-III were increased in the livers of the BDL mice and then decreased after treatment with GGCLT (Figure 5). GGCLT treatment also decreased the hepatic mRNA levels of VEGF, VEGFR1, and VEGFR2 in the BDL mice (Figure 6). GGCLT treatment suppressed the increase of HIF-1α and fibrogenesis- and angiogenesis-related factors in the livers of BDL-challenged mice.

## 3. Discussion

Hypoxia-inducible factors are key mediators of fibrogenesis and angiogenesis in hypoxic liver cells. Therefore, a pharmaceutical strategy for liver fibrosis involving inhibition of the level of HIF-1α may be beneficial for patients with liver disease. It has been reported that HIF-1α is activated in mice with BDL-induced liver fibrosis [23]. Despite the relatively high mortality rates due to bile leakage and rupture of biliary cysts, BDL-induced liver fibrosis is still widely used in mice models. This is because of its advantage of more convenient manipulation on targeted gene knockouts, which is a powerful strategy to address genic disorders. Our findings demonstrated that GGCLT treatment improved liver fibrosis caused by BDL in mice, in part through a decrease in HIF-1α-induced elevation of inflammation, oxidative stress, and angiogenesis-related factors.

Hydroxyproline serves as a marker of collagen deposition in liver fibrosis. According to a report by Moczydlowska et al., BDL results in the significant upregulation of hydroxyproline, HIF-1α, TGF-β, MMP-2, and MMP-9 in rats with liver fibrosis [22]. Moreover, Copple et al. suggested that HIF-1α deficiency is responsible for the reduction of collagen deposition in the liver of mice after BDL challenge as it reduces hepatic α-SMA and type I collagen expression [25]. It is noteworthy that the transcriptional activity of NF-κB is critical for HIF-1α to contribute to hypoxic liver disease [4,39]. In the present study, we observed a significant reduction in plasma ALT levels, inflammatory cell infiltration, and hepatic levels of hydroxyproline, NF-κB, α-SMA, TGF-β, MMP-2 and MMP-9 in the BDL mice after treatment with 300 mg/kg of GGCLT. Therefore, for our BDL-challenge mouse model, we speculate that hepatic inflammatory cells may have triggered HSCs to produce a deposition of collagen by releasing profibrotic growth factors, thus causing liver fibrosis, and GGCLT diminished these phenomena. Meanwhile, we suggest that NF-κB plays an essential role in these mechanisms. Of note, portal fibroblasts have a dominant role—alongside HSCs—in early liver fibrogenesis in rats through their stimulation of α-SMA levels and ECM deposition during the first 72 h after BDL challenge [20,21]. In addition, portal myofibroblasts promote angiogenic properties and vascular remodeling by increasing VEGFA and VEGFR2 levels, thereby underlying hepatic fibrosis formation [8]. However, there has been little exploration of the relationship between proliferation of portal fibroblasts and a hypoxic microenvironment (e.g., upregulation of HIF-1α) at the beginning of cholestatic liver fibrosis. Therefore, our future studies on whether GGCLT treatment could restrain expansion of portal fibroblasts and enhance HIF-1α in the liver in the early stages after a BDL surgical procedure (e.g., 1 to 3 days) are warranted.

VEGF is a crucial angiogenic factor that enhances wound healing and tumor growth in hypoxia. It has distinctive binding specificities to VEGFR1 and VEGFR2, which are expressed in the vascular endothelium to mediate its biologic effects [40]. Moreover, VEGF and angiopoietin-1, produced by activated HSCs, have been shown to trigger angiogenesis in a murine model with liver fibrosis or following exposure to leptin [41,42]. Based on our findings, we suggest that VEGF enhances angiogenesis through VEGFR-1 and VEGFR-2 in the BDL mouse model, in which GGCLT inhibited VEGF.

In rats, BDL increases oxidative stress in the liver, which is characterized by increases in hepatic lipid peroxidation and total nonprotein mixed disulfides, as well as a reduction in glutathione [43,44]. Of note, oxidative stress is a major feature of experimental models of fibrosis. Antioxidants have been proven to be partially effective in the treatment of fibrosis and cirrhosis [45]. In our study, the hepatic hydroxyproline and malondialdehyde levels were obviously increased in experimental cholestatic liver fibrosis, revealing increased oxidative stress. In contrast, BDL-challenged mice treated with GGCLT displayed a vigorous increase in the level of hepatic GSH. Thus, GGCLT may act as an antioxidant. GGCLT is widely reported to ameliorate gastrointestinal dysfunction and upregulate the immune response to combat inflammation, and it has been beneficial for the treatment of infectious diseases [46]. However, the mechanism underlying GGCLT’s antifibrotic activity is still unclear. Based on our findings, we suggest that the possible mechanisms of the anti-liver fibrosis effect of GGCLT (i.e., suppression of hepatic oxidative stress and angiogenesis) may be dependent on an HIF-1α-mediated pathway. As such, a schematic hypothesis of the anti-liver fibrosis effect of GGCLT is shown in Figure 7. Taken together, cholestatic cholangiopathy and liver fibrosis, which result from biliary duct obstruction, may manifest early in life and present a therapeutic challenge that necessitates surgical operation. According to the principles of treatment in traditional Chinese medicine (TCM), GGCLT aims to "clear heat and detoxify", and may be used to treat liver fibrosis. Therefore, we suggest GGCLT as a complementary therapeutic strategy in chronic liver disease for clinical practice.

## 4. Materials and Methods

### 4.1. Preparation of GGCLT

The preparation and formula analysis of GGCLT were conducted according to our previous study [31]. Powdered constituents of GGCLT were mixed from the crude ingredients of *P. radix*, *S. radix*, *C. rhizome*, and *G. radix* at a ratio of 8:3:3:2 by weight. Firstly, the GGCLT power was boiled with dH_2_O at 80 °C for 2 h. The mixture was filtered and frozen at −80 °C to create a paste crystal residue, with an approximate yield of 12.5% *(w/w)*. These contents were lyophilized at −20 °C. The dried compound was subsequently dissolved in dH_2_O before use. 

### 4.2. Animals and Experimental Protocols

Male C57BL/6 mice (5–6 weeks old) were obtained from the National Laboratory Animal Center (Taipei, Taiwan) and were housed in a temperature-controlled chamber at 20 ± 2 °C under a 12 h light/dark cycle. Mice were randomly grouped and treated with either dH_2_O or GGCLT at the three doses of 30, 100, and 300 mg/kg body weight. Treatment was administered via gastric gavage, occurring once per day for 4 weeks. Hepatic fibrosis in mice was performed through a BDL procedure. In brief, mice were anaesthetized with inhalation of 4 vol% isoflurane in 100% oxygen at a flow rate of 4 L/min for execution of a ventral laparotomy. Subsequently, the common bile duct above the duodenum was ligated with 3-0 silk by two ligatures and sectioned between the ligatures. The midline abdominal incision was closed with catgut. The sham-control mice also underwent a laparotomy and manipulation of the common bile duct but without ligation. In addition, the sham-control mice were gavaged with an equal volume of dH_2_O. At the end of the procedure, the mice were anesthetized by CO_2_ inhalation prior to decapitation. The plasma and liver tissues were collected for further analysis. All experimental animal protocols were approved by the Chang Gung Memorial Hospital Animal Care and Use Committee (IACUC No.2016060603) in accordance with the International Guidelines for Humane Animal Care and Use.

### 4.3. Analyses of Liver Histology, Cytokines, and Biochemicals

Liver hematoxylin and eosin staining was performed according to our previous protocol [31]. Plasma alanine aminotransferase was measured using a commercially available colorimetric kit (Randox Laboratories, Antrim, UK). The hepatic hydroxyproline content was determined according to previously published methods [31]. The production of thiobarbituric acid reactive substances (TBARS) was used to represent the amount of lipid peroxidation and was expressed as malondialdehyde equivalents. The amount of malondialdehyde was assessed by spectrophotometric measurement of the absorbance at 535 nm and 37 °C; the results are expressed as nmol/mg protein. The hepatic glutathione was analyzed colorimetrically using a commercially available assay kit (Cayman, MI, USA) in accordance with the manufacturer’s instructions. 

### 4.4. Quantitative Real-Time PCR 

Total RNA was prepared using the guanidinium-phenol-chloroform method. Reverse transcription was performed using a RevertAid First Strand cDNA Synthesis Kit (Thermo Scientific, Waltham, MA, USA) in accordance with the manufacturer’s protocol. Quantitative real-time PCR using the SYBR system was conducted on a LightCycler 1.5 apparatus (both from Roche Applied Science, Mannheim, Germany). The PCR reaction was performed under the following conditions: 95 °C for 10 min and then 45 cycles of 95 °C for 15 s, 57 °C for 30 s and 72 °C for 30 s. The real-time PCR data were normalized to glyceraldehyde 3-phosphate dehydrogenase (GAPDH) expression. The primer sequences are listed in Table 1.

### 4.5. Western Blot Measurement

The liver tissue was lysed in 0.5 mL of CelLytic M lysis reagent (Sigma-Aldrich, St. Louis, MO, USA) with 1% phosphatase inhibitor cocktail and protease inhibitor cocktail and centrifuged at 13,000× *g* for 30 min at 4 °C. The concentration of protein lysate was determined using the Bradford assay. In addition, the nuclear fraction was harvested using a nuclear extraction kit (Abcam, Cambridge, MA, USA) in accordance with the manufacturer’s instructions. The cell lysates were separated using sodium dodecyl sulfate (SDS)-polyacrylamide gel electrophoresis and transferred onto polyvinylidene fluoride (PVDF) membranes, following incubation with primary antibodies against NF-κB, α-SMA, TGFβ-R1, and TGFβ-R2 (all from Santa Cruz Biotechnology, Dallas, TX, USA). Finally, horseradish peroxidase-conjugated secondary antibodies were added, and the reaction was detected by electrochemiluminescence. The data were calibrated using H1 and β-actin as internal controls.

### 4.6. Statistical Analysis

All samples for each experimental analysis were randomized. Data are expressed as a mean ± SEM. Quantitative variables were compared using Student’s *t*-tests. *p* < 0.05 was considered statistically significant.

## Figures and Tables

**Figure 1 ijms-20-04181-f001:**
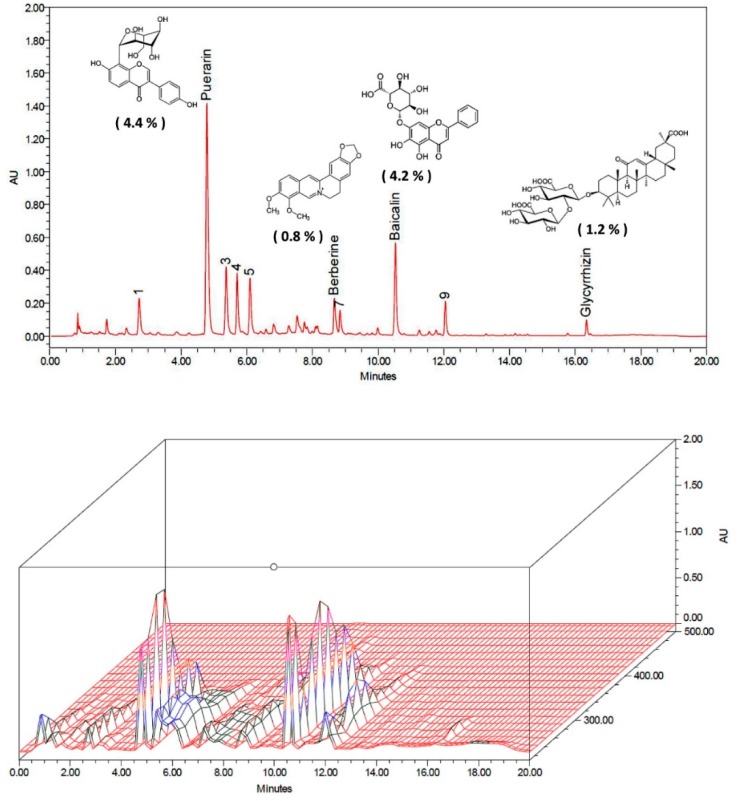
Chromatogram of Ger-Gen-Chyn-Lian-Tang (GGCLT) by high-performance liquid chromatography (HPLC) analysis. The upper panel shows the peaks, compound structure, and ingredient percentages of puerarin, baicalin, berberine, and glycyrrhiza, the four main index components of *Puerariae radix, Scutellariae radix, Coptisis rhizoma and Glycyrrhizae radix*, respectively. The lower panel shows the fingerprint of the three-dimensional structures of these four main index components, as determined by HPLC analysis.

**Figure 2 ijms-20-04181-f002:**
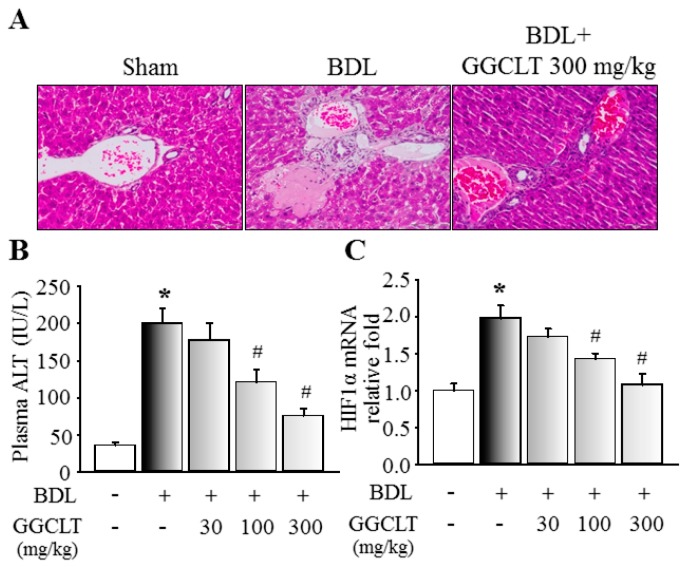
GGCLT-attenuated bile duct ligation (BDL)-induced liver injury in mice. (**A**) Representative histological results of liver tissue stained with hematoxylin and eosin from a sham-control mouse, a BDL mouse, and a BDL mouse treated with 300 mg/kg GGCLT (200× magnification); (**B**) plasma alanine aminotransferase (ALT) levels; and (**C**) hepatic hypoxia-inducible factor-1α (HIF-1α) mRNA levels. * *p* < 0.05 vs. sham-control mice (*n* = 6), ^#^
*p* < 0.05 vs. BDL mice (*n* = 8).

**Figure 3 ijms-20-04181-f003:**
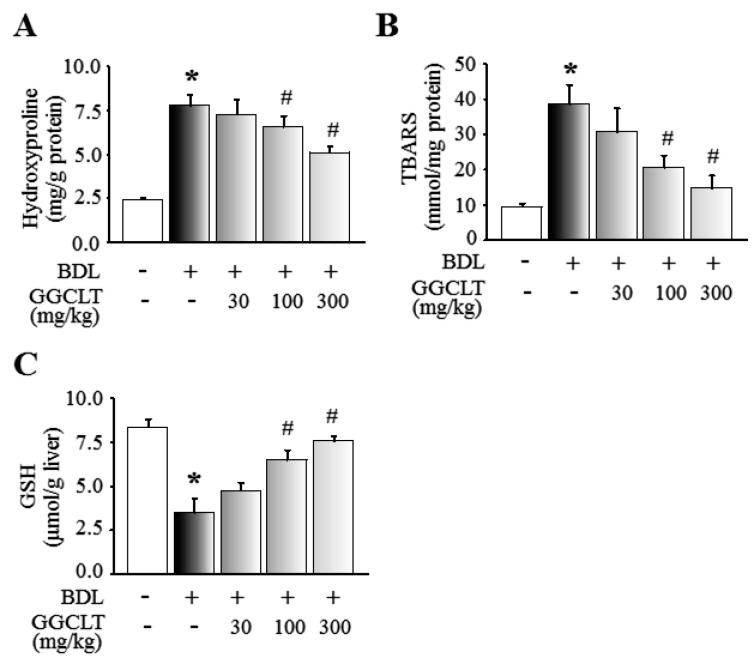
GGCLT reduced BDL-induced oxidative stress in the liver tissues of mice. (**A**) Hepatic hydroxyproline content; (**B**) hepatic lipid peroxidation represented by thiobarbituric acid reactive substances (TBARS); and (**C**) hepatic glutathione (GSH) levels. * *p* < 0.05 vs. sham-control mice (*n* = 6); ^#^
*p* < 0.05 vs. BDL mice (*n* = 8).

**Figure 4 ijms-20-04181-f004:**
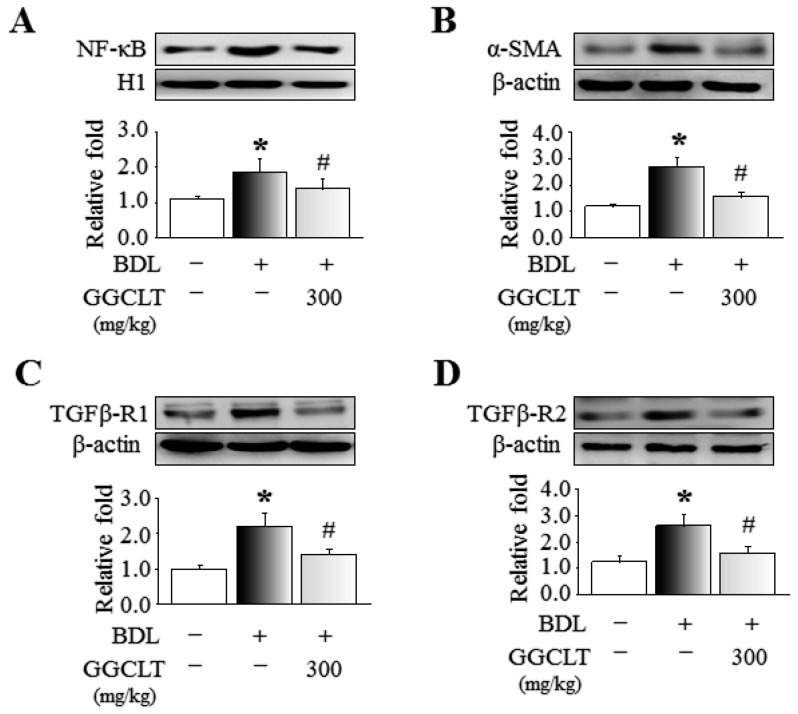
Western blot analysis of hepatic protein levels of (**A**) nuclear factor kappa B (NF-κB), (**B**) α-smooth muscle actin (α-SMA), (**C**) transforming growth factor-β (TGFβ)-R1 and (**D**) TGFβ-R2 in the liver tissues of sham-control mice and BDL mice treated with or without 300 mg/kg GGCLT. The data presented are the mean ± SEM from three independent measurements. * *p* < 0.05 vs. sham-control mice; ^#^
*p* < 0.05 vs. BDL mice.

**Figure 5 ijms-20-04181-f005:**
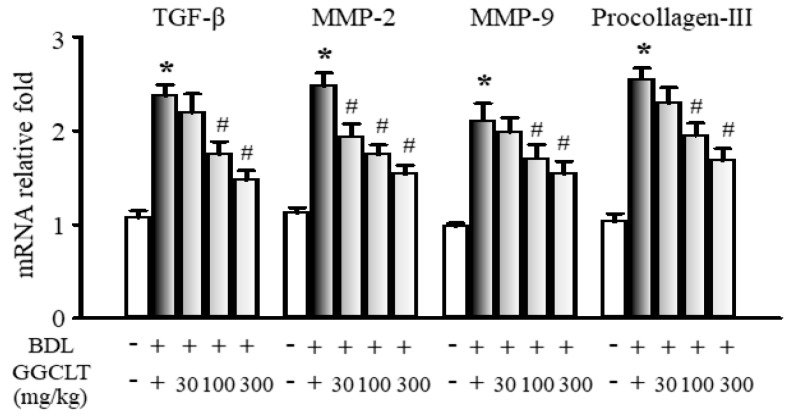
Effects of GGCLT on hepatic fibrotic signaling in BDL mice. The quantitative real-time PCR results show the mRNA levels of TGF-β, matrix metalloproteinase-2 (MMP-2), MMP-9, and procollagen-III in the liver tissues. The data presented are the mean ± SEM from three independent measurements. * *p* < 0.05 vs. sham-control mice; ^#^
*p* < 0.05 vs. BDL mice.

**Figure 6 ijms-20-04181-f006:**
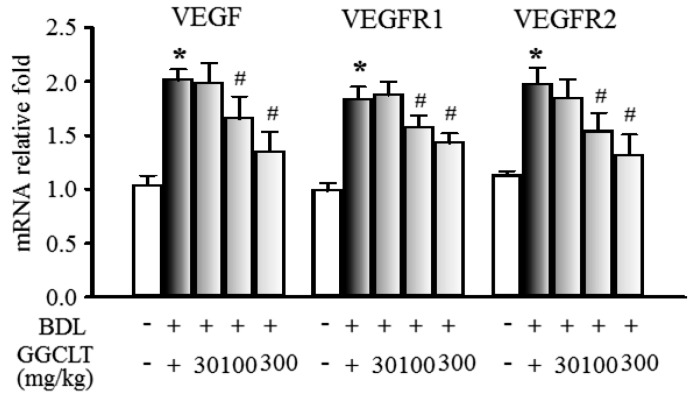
Effects of GGCLT on hepatic levels of HIF-1α and angiogenesis-regulated factors in BDL mice. The quantitative real-time PCR results show the mRNA levels of vascular endothelial growth factor (VEGF), VEGFR1, and VEGFR2 in the liver tissues. The data presented are the mean ± SEM from three independent measurements. * *p* < 0.05 vs. sham-control mice; ^#^
*p* < 0.05 vs. BDL mice.

**Figure 7 ijms-20-04181-f007:**
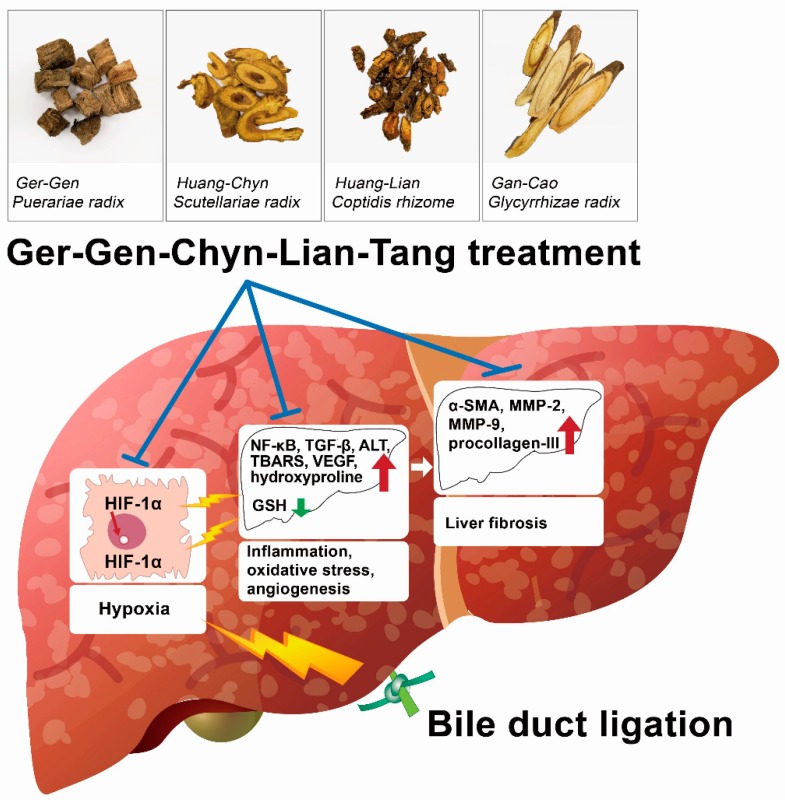
Schematic representative hypothesis of the use of GGCLT treatment to decrease inflammation, oxidative stress and angiogenesis, and further improve liver fibrosis in BDL-challenged mice in our study.

**Table 1 ijms-20-04181-t001:** Primer sequences used for quantitative real-time PCR analysis.

Gene	Forward	Reverse
HIF-1α	TCAAGTCAGCAACGTGGAAG	TATCGAGGCTGTGTCGACTG
VEGF	GAGAGAGGCCGAAGTCCTTT	TTGGAACCGGCATCTTTATC
VEGFR1	GAAGCGGTTCACCTGGACTGAGACC	GGCTTTGCTGGGGGGATTTCTCTAA
VEGFR2	ACAGCAGTGGGATGGTCCTTGCAT	AAACAGGAGGTGAGCTGCAGTGTGG
TGF-β	TGCCCTCTACAACCAACACAACCCG	AACTGCTCCACCTTGGGCTTGCGAC
MMP-2	GCTGATACTGACA CTGGTACTG	CAATCTTTTCTGGGAGCTC
MMP-9	CGTCGTGATCCCCACTTACT	AGAGTACTGCTTGCCCAGGA
Procollagen-III	CCCCTGGTCCCTGCTGTGG	GAGGCCCGGCTGGAAAGAA
GAPDH	CCCTTCATTGACCTCAACTACATGG	CATGGTGGTGAAGACGCCAG

HIF-1α, hypoxia-inducible factor-1α; VEGF, vascular endothelial growth factor; VEGFR1 and 2, vascular endothelial growth factor receptor 1 and 2; TGF-β, transforming growth factor-β; MMP-2 and 9, matrix metalloproteinase-2 and 9.

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
