# Peer review of "Positive Effects of Ger-Gen-Chyn-Lian-Tang on Cholestatic Liver Fibrosis in Bile Duct Ligation-Challenged Mice"

_ijms, 2019, doi:10.3390/ijms20174181_

Round 1

Reviewer 1 Report

Revised version is now ok. 

Author Response

Response to Reviewer 1 comments

Point: Revised version is now ok.

Response:

We are very grateful to reviewer to recommend our article for publication.

Reviewer 2 Report

The study: The effects of Ger-Gen-Chyn-lian-Tang on cholestatic liver fibrosis in bile duct ligation-challenged mice is well-designed study. The results clearly show the impact of  Ger-Gen-Chyn-lian-Tang  on fibrosis in liver. It would be beneficial if the authors describe BDL mice in details. For now, I would request this change to be added before publishing.

Author Response

Response to Reviewer 2 comments

Point: The study: The effects of Ger-Gen-Chyn-lian-Tang on cholestatic liver fibrosis in bile duct ligation-challenged mice is well-designed study. The results clearly show the impact of Ger-Gen-Chyn-lian-Tang on fibrosis in liver. It would be beneficial if the authors describe BDL mice in details. For now, I would request this change to be added before publishing.

Response:

We thank the reviewer for drawing our attention to this point. We have added the information in the Introduction section (page 5, line 80 to 84 and page 5, line 86 to 89), the Discussion section (page 8, line 159 to 162) and the Materials and Methods section (page 11, line 233 to 239) according to the reviewer’s suggestion and described as follow:

Bile duct ligation (BDL) is a surgical procedure in the common bile duct that is used to cause extrahepatic biliary obstruction which mimics cholestatic liver damage. The process of BDL-induced liver cirrhosis is that it induces the rapid growth of biliary epithelial cells and oval cells, then causes bile ductular proliferation, cholestasis, and fibrosis, and ultimately leads to secondary biliary cirrhosis (Geerts et al., 2008; Kountouras et al., 1984; Popov et al., 2010). (In the Introduction section, page 5, line 80 to 84).

In this regard, BDL in mice is a widespread model used to induce biliary inflammation, fibrosis and cholestatic liver injury (Prado et al., 2003). In addition, persinusoidal- and periportal fibrosis have obviously developed on day 10 and day 20, separately after BDL surgery in mice (Tag et al., 2015). (In the Introduction section, page 5, line 86 to 89).

Despite the relatively high mortality rates due to bile leakage and rupture of a biliary cyst, the BDL-induced liver fibrosis is still widely used in mice model for its advantage of more convenient manipulation and on targeted gene knockouts which is a powerful strategy to address the genic disorders. (In the Discussion section, page 8, line 159 to 162).

Hepatic fibrosis in mice was performed through BDL procedure. In brief, mice were anaesthetized with inhalation of 4 vol% isoflurane in 100 % oxygen at a flow rate of 4 L/min and executed a ventral laparotomy. Subsequently, the common bile duct above the duodenum was ligated with 3-0 silk by two ligatures and sectioned between the ligatures. The midline abdominal incision was closed with catgut. The sham-control mice were performed by laparotomy and manipulation of the common bile duct but without ligation. In addition, the sham-control mice were gavaged with an equal volume of dH2O. (in the Materials and Methods section, page 11, line 233 to 239)

Reference:

Geerts, A.M.; Vanheule, E.; Praet, M.; Van Vlierberghe, H.; De Vos, M.; Colle, I.  Comparison of three research models of portal hypertension in mice: macroscopic, histological and portal pressure evaluation. Int. J. Exp. Pathol. 2008, 89, 251-263.

Kountouras, J.; Billing, B.H.; Scheuer, P.J. Prolonged bile duct obstruction: a new experimental model for cirrhosis in the rat. Br. J. Exp. Pathol. 1984, 65, 305-311.

Popov, Y., Sverdlov, D.Y., Bhaskar, K.R., Sharma, A.K., Millonig, G., Patsenker, E., Krahenbuhl, S.; Krahenbuhl, L.; Schuppan, D. Macrophage-mediated phagocytosis of apoptotic cholangiocytes contributes to reversal of experimental biliary fibrosis. Am. J. Physiol. Gastrointest. Liver Physiol. 2010, 298, G323-334.

Prado, I.B.; Santos, M.H.H.; Lopasso, F.P.; Iriya, K.; Laudanna, A.A. Cholestasis in a murine experimental model: lesions include hepatocyte ischemic necrosis. Rev. Hosp. Clín. 2003, 58, 27-32.

Tag, C.G.; Sauer-Lehnen, S.; Weiskirchen, S.; Borkham-Kamphorst, E.; Tolba, R.H.; Tacke, F.; Weiskirchen, R. Bile duct ligation in mice: induction of inflammatory liver injury and fibrosis by obstructive cholestasis. J. Vis. Exp. 2015, e52438.

Round 2

Reviewer 2 Report

The manuscript in its current form is ready for submission. The authors addressed all concerns appropriately.

This manuscript is a resubmission of an earlier submission. The following is a list of the peer review reports and author responses from that submission.

Round 1

Reviewer 1 Report

The paper aims to investigate whether Ger-Gen-Chyn-Lian-Tang, an officially standardized mixture of Chinese herbal medicines, suppress  oxidative stress, inflammation, and angiogenesis during experimental liver fibrosis.

Authors assert that HIF-1α is involving in the pathways underlying cellular and molecular mechanisms of hepatic fibrogenesis

I have few comments:

1) I think that the authors should provide to the readers, some information about the biologically active compounds from Ger-Gen-Chyn-Lian-Tang, underlying whether they should be involved in the pharmacologic molecular mechanism.

2) In the papers authors give evidence that the mRNA levels of HIF-1α decreased in response to treatment with Ger-Gen-Chyn-Lian-Tang;  moreover they affirm that the administration of GGCLT suppressed the induction of hepatic hydroxyproline. Therefore I think that the authors should clarify if Ger-Gen-Chyn-Lian-Tang suppress the expression of HIF-1α through a transcriptional mechanism that leads to low levels of HIF-1α mRNA or through a molecular mechanism, dependent from the suppressed induction of hydroxyproline that leads to a reduced stabilization of HIF-1α.

Author Response

 Sincerest thanks to the reviewer for the time and comments on our manuscript. We have addressed all issues indicated in the review report at point by point, and believed that the revised version may meet the reviewers requirements.

Reviewer 2 Report

This is a careful study. I have some points that should be considered:

Please add in the abstract actual values with SEM and p value of the most important parameters.

You mention that GGCLT is useful for treatment. Please clarify in the text whether this statement is  based on RCTs.

In the discussion section, please include a para to demonstrate why your study is of importance in a clinical setting. 

Author Response

(The authors gave the same response as above.)
